# Comprehensive analysis of cancer breakpoints reveals signatures of genetic and epigenetic contribution to cancer genome rearrangements

**Kseniia Cheloshkina[1,2], Maria Poptsova[1]***

**1** Laboratory of Bioinformatics, Faculty of Computer Science, National Research University Higher School of Economics, Moscow, Russia, **2** Faculty of Digital Transformation, ITMO University, St. Petersburg, Russia

* mpoptsova@hse.ru

**Data Availability Statement:** All relevant data are within the manuscript and its Supporting Information files or available at the following url:

## Abstract

Understanding mechanisms of cancer breakpoint mutagenesis is a difficult task and predictive models of cancer breakpoint formation have to this time failed to achieve even moderate predictive power. Here we take advantage of a machine learning approach that can gather important features from big data and quantify contribution of different factors. We performed comprehensive analysis of almost 630,000 cancer breakpoints and quantified the contribution of genomic and epigenomic features–non-B DNA structures, chromatin organization, transcription factor binding sites and epigenetic markers. The results showed that transcription and formation of non-B DNA structures are two major processes responsible for cancer genome fragility. Epigenetic factors, such as chromatin organization in TADs, open/closed regions, DNA methylation, histone marks are less informative but do make their contribution. As a general trend, individual features inside the groups show a relatively high contribution of G-quadruplexes and repeats and CTCF, GABPA, RXRA, SP1, MAX and NR2F2 transcription factors. Overall, the cancer breakpoint landscape can be represented by well-predicted hotspots and poorly predicted individual breakpoints scattered across genomes. We demonstrated that hotspot mutagenesis has genomic and epigenomic factors, and not all individual cancer breakpoints are just random noise but have a definite mutation signature. Besides we found a long-range action of some features on breakpoint mutagenesis. Combining omics data, cancer-specific individual feature importance and adding the distant to local features, predictive models for cancer breakpoint formation achieved 70–90% ROC AUC for different cancer types; however precision remained low at 2% and the recall did not exceed 50%. On the one hand, the power of models strongly correlates with the size of available cancer breakpoint and epigenomic data, and on the other hand finding strong determinants of cancer breakpoint formation still remains a challenge. The strength of predictive signals of each group and of each feature inside a group can be converted into cancer-specific breakpoint mutation signatures. Overall our results add to the understanding of cancer genome rearrangement processes.

https://github.com/KseniiaCheloshkina/cancer_breakpoints_hotspots_prediction_wide.

**Funding:** K.C. and M.P were supported by the Centre of Fundamental Research of the National Research University Higher School of Economics (https://cfi.hse.ru/). The funder had no role in study design, data collection and analysis, decision to publish, or preparation of the manuscript.

**Competing interests:** The authors have declared that no competing interests exist.

## Author summary

We analysed more than half a million breakpoints from all major cancer types and quantified contributions of genetic and epigenetic factors to cancer breakpoint mutagenesis. The results suggest that transcription and formation of non-B DNA structures are the two major processes responsible for cancer genome fragility. Epigenetic factors, such as chromatin organization in TADs, open/closed regions, histone marks are less informative while still contributive. Despite the common trends, each cancer type has its own peculiarities. Breakpoint hotspots in brain can be predicted by distribution of non-B DNA structures, those in liver by transcription factor binding sites, those in blood by non-B DNA structures and promoter regions. Cancer breakpoint landscape can be viewed as hotspots and individual breakpoints scattered all over the genome. Hotspots have distinct genomic and epigenomic signatures with relative contribution varied for different cancer types. Individual cancer breakpoints are the mixture of random noise and breakpoints with a recognizable mutation signature. Quantifying contribution of different factors to cancer breakpoint mutagenesis for individual cancer genomes will enhance our understanding of individual mechanisms of cancer genome rearrangement.

## Introduction

Cancer genomes are unstable and undergo numerous rearrangements resulting in origination of structural variants such as deletions, insertions, translocations, and copy number variants. Over the last 20 years several consortium cancer genome projects–The Cancer Genome Atlas (TCGA) [1], International Cancer Genome Consortium (ICGC) [2], and the ICGC/TCGA Pan-Cancer Analysis of Whole Genomes (PCAWG) Project [3]–published the information on point and structural mutations in thousands of common and rare cancer genomes. Identifying cancer mutation determinants is extremely important for understanding the genomics of the disease, but the heterogeneity of cancer genome mutations presents major difficulties in the analysis of cancer genomes.

Employing a machine learning approach helped better understand the determinants of cancer point mutations at 1Mb scale [4,5] The density of the histone mark H3K9me3, which is associated with heterochromatin, explained 40% of the variance of cancer point mutation densities [4]. The machine learning model built on chromatin accessibility (via DNase I hypersensitive sites), histone modifications and replication timing together reached R2 of 86%, and the most important features for each cancer type were those from the cell of origin [5]. Other studies pointed to other factors such as DNA mismatch repair state (measured via microsatellite instability status) [6], CTCF binding sites [7], DNA wrapped around nucleosomes, or transcription factors bound to DNA (reviewed in [8]), but the contribution of each factor separately or jointly was not assessed. In study [9] the authors demonstrated that non-B DNA structures, such as G-quadruplexes, triplexes, Z-DNA, cruciforms, direct and inverted repeats can explain 37% (breast) to 52% (malignant lymphoma) of cancer point mutation 0.5 Mb densities. Adding histone modifications could increase prediction power of the models by 10–15%, but even the best model did not exceed R2 of 76%.

Determining the most influential factors which influence cancer breakpoints appears to be a more difficult task than that for point mutations. Statistical enrichment approach showed association of many genomic features with cancer breakpoints, but machine learning models for predicting cancer breakpoint densities using the same predictors as for the point mutation

models did not achieve the same predictive power. Different studies showed enrichment of cancer breakpoint regions with non-B DNA structures [9–13]. However predicting breakpoint densities with the same set of non-B DNA structures and epigenetic factors (as it was done for cancer point mutations) could only achieve R2 of 10% for all cancers with the exception of 18% for breast cancer. [9]. When studying the differential impact of only two types of non-B DNA structures, quadruplexes and stem-loops, machine learning models predicting breakpoint hotspots did not exceed 60% median ROC AUC for all cancer types except bone (66%) and breast (70%) [14]. The summary of all mentioned cancer-related studies is given in S1 Table.

In the present study we took advantage of the available omics data and built models that would aggregate information from different levels of genome organization. Using the machine learning approach, we systematically explored the contribution of different groups of genomic and epigenomic factors–non-B DNA structures, genomic regions, chromatin structure, transcription factor binding sites and epigenetic markers–to cancer breakpoint region formation. We also tested and compared different approaches that use local and/or distant feature densities and/or presence/absence of factors in the vicinity of breakpoints. In addition, we explored the degree of randomness of breakpoints and its effect on prediction power of machine learning models. Selecting the most important features at the individual level of cancer genome, for the first time we could build a model that improved on all of the previous machine learning models for cancer breakpoint prediction. However the task of cancer breakpoint prediction still remains a challenge.

## Results

We used cancer breakpoint data available at the International Cancer Genome Consortium (ICGC) Data Portal. The data consist of 2,803 samples covering 10 cancer types with 652,586 breakpoints in total. In general, 1,588–197,700 breakpoints from 16–646 donors are provided for each cancer type with breast and brain cancer comprising maximal and minimal number of breakpoints, respectively. For modelling we calculated breakpoint densities at 100 kb non-overlapping windows and defined hotspots in high-density regions at three different thresholds (see Methods and S2–S4 Tables). For model building we selected different genomic and epigenomic features that we combined into seven groups: non-B DNA structures (quadruplexes, Z-DNA, stem-loops, repeats) [15–17], histone modifications (HMs) [18], DNA methylation [18], transcription factor (TF) binding sites [18], chromatin accessibility (HDNase) [18], chromatin partitioning with topologically associated domains (TADs) [19], and genomic positions (whole genes, exons, introns, 5'- and 3'-UTRs promoters, downstream areas) (S1 Fig and S5 and S6 Tables and S1 File).

Here and further on throughout the modeling we are facing a class imbalance problem for which PR AUC is a more informative metric than ROC AUC. Even though we managed to reach a relatively high ROC AUC, the precision and recall remained low (see Discussion). Nevertheless, the lift of recall and lift of precision (which shows how many times a model is better than random guessing) assume the values of 2–10, which points to the presence of certain regularities. Since the main focus of the paper is to examine the contribution of features to the predictive power of the model and to verify if omics approach helps to improve an ML model prediction power, we used ROC AUC and lift of recall to compare the performance of models built with features and different groups of features.

### Contribution of each group of features to cancer breakpoint prediction

We built machine learning models to investigate the impact of every feature group on prediction of breakpoint hotspots (S7 Table) by modeling breakpoint hotspots based on each feature

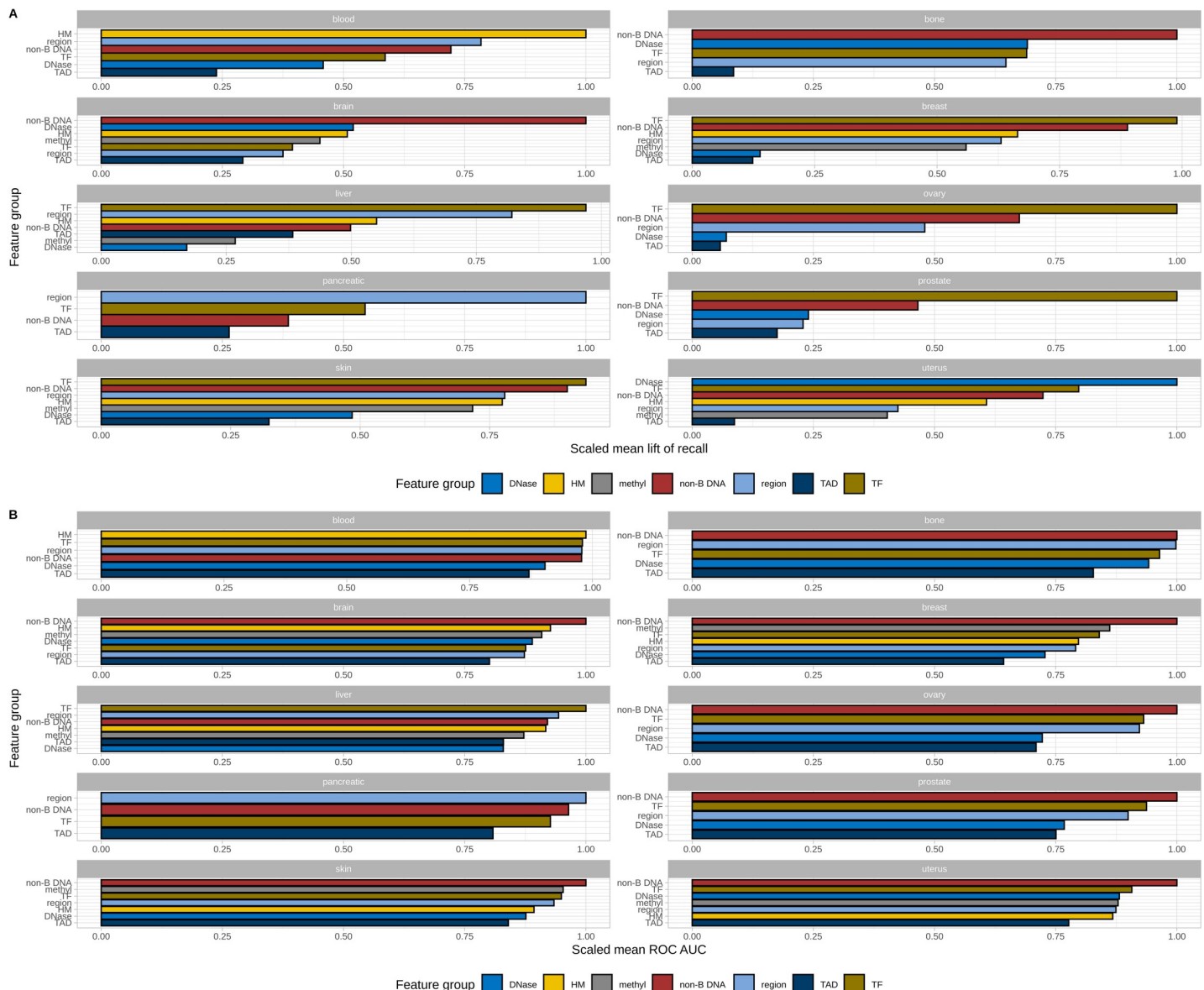

**Fig 1. Feature group ranging by model performance using two different metrics–lift of recall and ROC AUC. A.** Mean lift of recall for 0.03 probability percentile for each cancer type and feature group scaled and averaged for 99% and 99.5% labelling types. **B.** Mean ROC AUC for each cancer type and feature group scaled and averaged for 99% and 99.5% labelling types.

group. Each feature group was ranged according to the model performance (see Methods), and the results are presented in Fig 1A. The results show that the best feature group significantly (by 0.25) outperforms others for almost all cancer types, and this value even reaches 0.5 for 3 cancers (blood, pancreas and prostate). This outperforming group included TFs for 5 cancers (liver, skin, prostate, ovary, breast), non-B DNA structures for 2 cancers (brain and bone), genomic regions (pancreas), HMs (blood), and HDNase (blood).

Non-B DNA structures always enter the top 3 important groups for all types of cancers, TFs are present in the top3 list in 8 types of cancer (except for blood and brain), genomic regions– in 5, and HMs in 4 cancer types. A similar ranging is observed when using ROC AUC (Fig 1B) as the model evaluation metric: non-B DNA structures represent the first important group in 7

cancer types and rank among the top-3 most important groups for all cancers. For the remaining three cancer types the most important groups are HMs (blood), genomic regions (pancreas), and TFs (liver). TFs ranked in the top-3 list for 9 cancers, genomic regions for 6 cancers, and methylation for 3 cancers (brain, breast and skin).

When analyzing group feature importance for all cancers combined, the non-B DNA structures are in the first place followed by transcription factors and genomic regions (S2A and S2B Fig) with the mean lift of recall (3.18 and 3.02) and the mean ROC AUC (0.66 and 0.62) slightly higher for non-B DNA than for TF. The results indicate that large-scale chromatin organization does not contribute much to the breakpoint formation: HDNAse and TADs occupy the last two places and HDNase is the one before last in the list. The two remaining in-between groups–HMs and methylation have nearly equal influence.

We also performed an analysis to define the best individual features inside the best feature groups–non-B DNA and TFs. We established that among all non-B DNA features G-quadruplexes and direct repeats are the most influential factors for all cancer types, while short tandem repeats and Z-DNA also make a valuable contribution to model performance (S3 Fig). Moreover, the importance of the best feature is several times higher than that of the remaining features emphasizing the key role of the selected candidate. Breakpoint hotspots locations on density coverage plots for 20Mb region on the chromosome 13 (S4 Fig) clearly demonstrate a correlation between cancer breakpoint hotspots and different non-B DNA features. Concerning transcription factors, sets of the most important features are less similar to each other for different cancer types but nevertheless CTCF, GABPA, RXRA, SP1, MAX and NR2F2 are more frequently included in top features than other transcription factors (S5 Fig). Interestingly, for the majority of cancer types distant features are of higher importance than the local features, and this is a common observation for two feature groups (see Results below).

Additionally, we investigated the influence of individual features with the Boruta feature selection method (see Methods) considering individual contributions of every feature from every group (S6 Fig and S8 Table). Totally, the method selected 50 important features with 5–23 features per cancer type (Fig 2). Similarly to the feature group importance results presented earlier, the Boruta selection method also highlighted features mostly from non-B DNA, TFs and genomic regions groups (at least one feature of each group is included in a set of best features for 8 cancer types). Nevertheless, all groups except for TADs contribute to the model prediction power.

Performing importance analysis on Boruta feature selection procedure results, we observed that features of only four feature groups (non-B DNA, TFs, genomic regions and HDNase) are included in the top list according to the criteria of a feature ranking important in at least 300 of 3000 analyzed datasets for all cancer types (see Methods and S7 Fig). In the resulting list the top five features are represented by direct repeats and G-quadruplexes, both local and distant (see Results below), together with distant binding sites of SP1 transcription factor. Other individual features that were selected less frequently include Z-DNA, short tandem repeats, mirror repeats, transcription factors RXRA, NR2F2, GABPA, CTCF, genomic regions such as 5' UTR, coding exons, 3' UTR, promoters and downstream areas, and also HDNase, but their influence differs for cancer types (S8 Fig). Interestingly, the majority of the features were captured at 1Mb scale, not at 100kB scale, revealing the importance of taking into account a wider landscape for cancer breakpoint hotspots identification (see Results below).

Independently from the methods, either group or Boruta feature importance,two best performing feature groups have invariably been those of non-B DNA and TFs following by genomic regions, of which 5'UTR and coding exons were the most contributing. This result emphasizes the key role of non-B DNA structures and transcription factors for cancer breakpoint hotspots prediction, however the contribution of individual features from these two

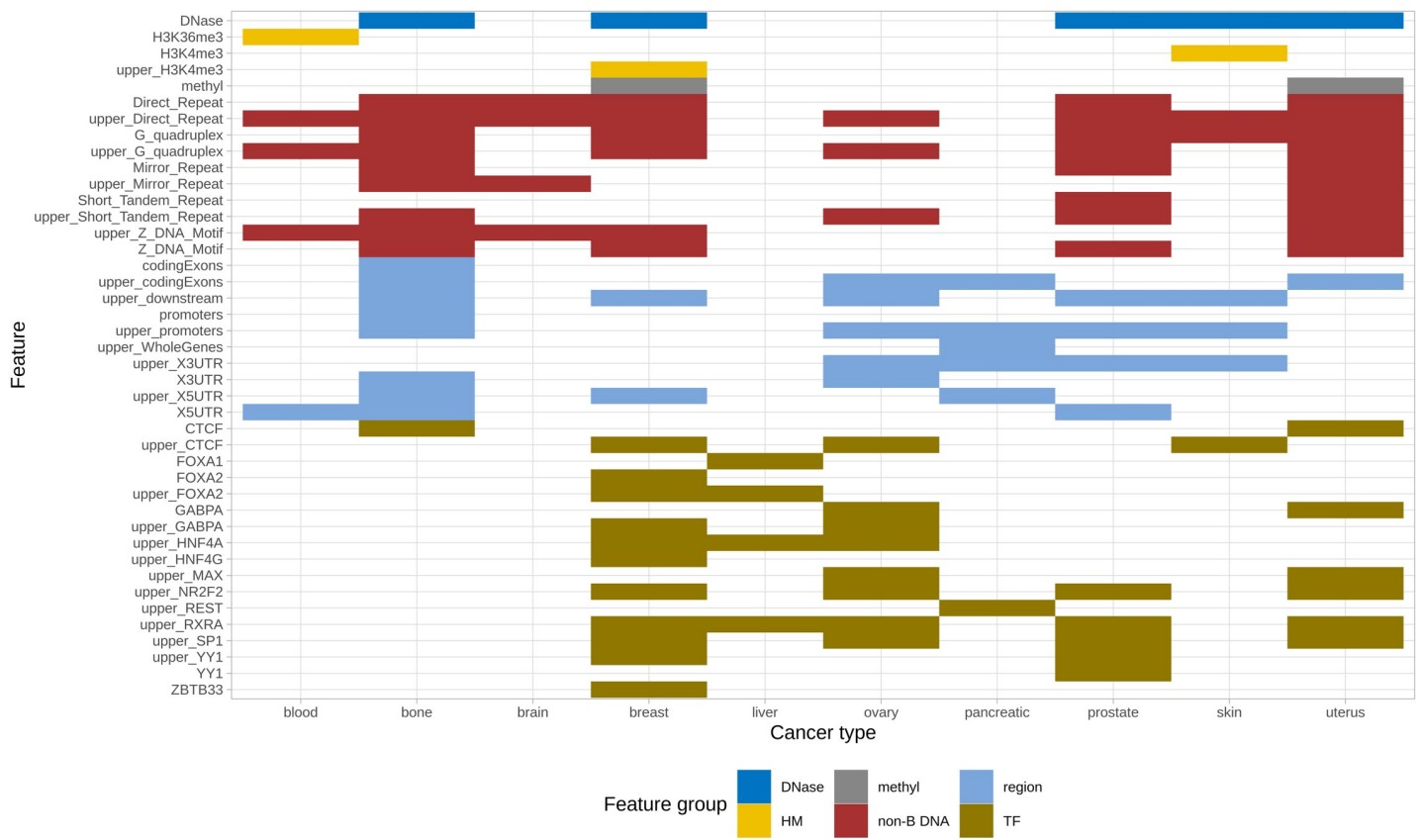

**Fig 2. Boruta selected feature sets by cancer type and feature group.** The most important features for all cancer types selected by Boruta feature selection method for each cancer type separately.

groups for each cancer type hotspots prediction is unequal. The importance of genomic regions such as 5'UTR and coding exons provides a hint about the major role of transcription processes in generating cancer breakpoints.

A comparison of model performance built on one group of features or on all features showed that an all-features model always outperforms the one-group-of-features model (either with the lift of recall or ROC AUC) with the exception of blood cancer (Figs 3 and S9 and S8 Table). We also report PR AUC as specifically informative for class-imbalance problems (Figs 3 and S9 and S8 Table). As we can see from Fig 3, according to PR AUC, the top three feature groups coincide with those determined according to ROC AUC or lift of recall for all types of cancer (except for the blood cancer for 99% labeling type), and the ranking of cancer types according to the model performances is generally preserved among three metrics. This result confirms the hypothesis that information from omics data improves the model prediction power [20]. However, it is of interest to note that the performance of the all-features model is only slightly higher than the performance of the model built on the best group of features. This result suggests the existence of structural variation signatures that would be very important to detect in the future, but is out of the scope of the present study.

To evaluate goodness-of-fit of the all-features and single feature group models, we calculated McFadden's R-squared (pseudo R-squared) for 99% and 99.5% labeling type (S10 Fig and S9 Table) by building logistic regression models. We found that for a half of cancers (breast, ovary, pancreas, skin, and blood) non-B DNA-based models produced the maximal

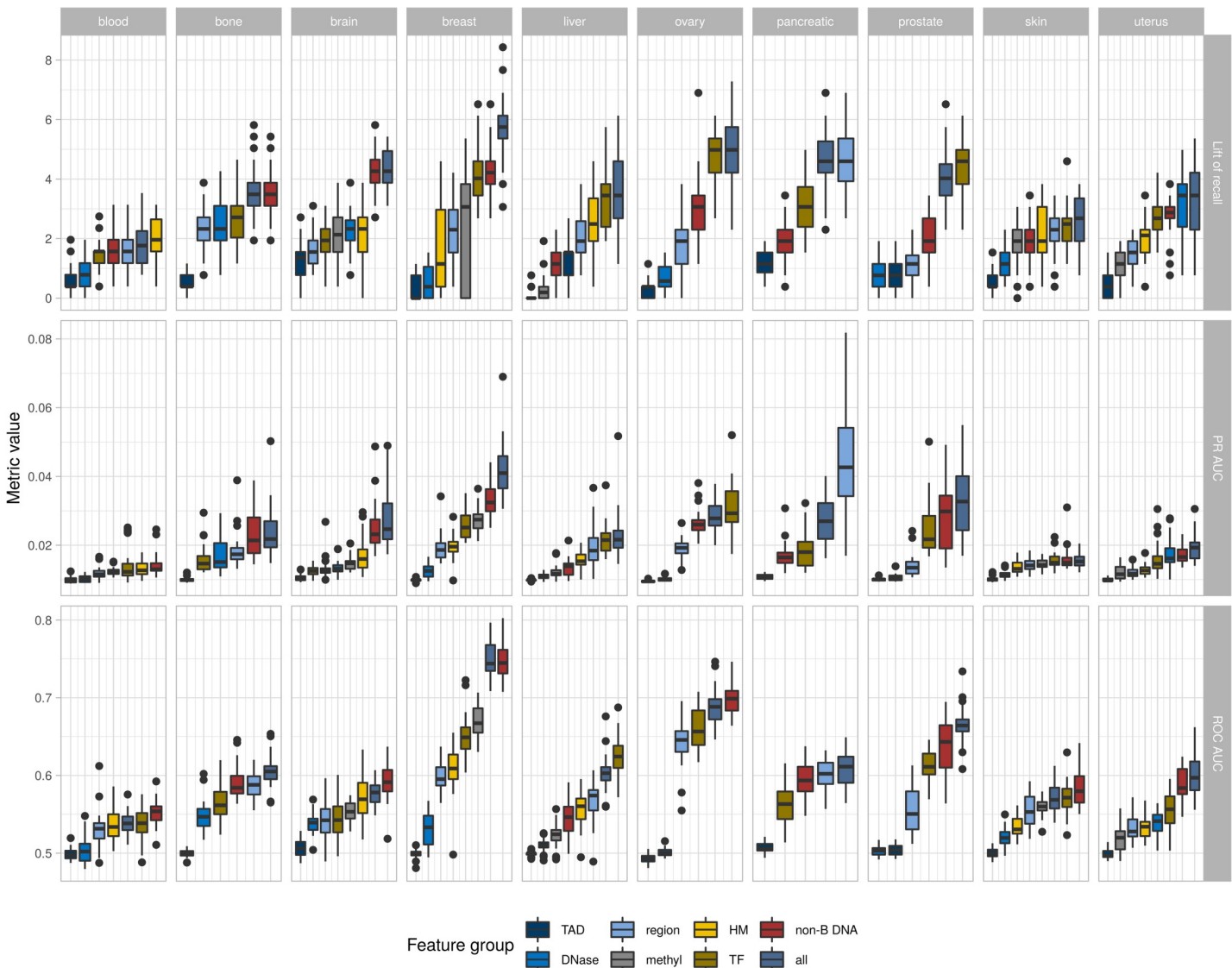

**Fig 3. Feature group contribution.** Contribution of different feature groups separately and all together. Distribution of the lift of recall (first panel) for 0.03 probability percentile, PR AUC (second panel) and ROC AUC (third panel) for each of one feature group-based models and models on Boruta selected features (for 99% labeling type).

pseudo R-squared while for the rest the all-features model performed the best. In absolute values, the maximal McFadden's R-squared over all cancer types are observed for breast, prostate and ovary cancers with 0.087, 0.066 and 0.061 respectively.

We investigated whether the different feature group models identify the same or different breakpoint hotspots regions. Towards that end we calculated the mean Spearman correlation between predictions for each pair of feature groups (S11A Fig). The highest mean correlation over all cancer types is observed for histone modification-based models that are positively associated with TF- and methylation-based models (with correlation of 0.47 and 0.46 respectively). On the opposite, TAD-based models generate the most specific predictions with maximal correlation of 0.045 with TF-based predictions. Secondly, for each genome window in each test set we calculated the number of hotspots predicted with different number (1, 2, or 3) of feature groups and analyzed the proportion of true hotspots. A comparison of median

quality of the best feature group model (corresponds to label "1") to overlapped predictions of 2- or 3-feature group models demonstrates that for the majority of cancer types a single feature group model is more precise than the overlapped predictions (S11B Fig). For several cancer types (uterus, breast, bone, pancreas) the median precision is only slightly higher with the increased standard deviation and the resulting number of identified hotspots (true positives) is lower. This fact, together with the observed uplift in combined models compared to the single feature group models, leads to a conclusion that in order to account for multiple factor contributions in cancer breakpoint hotspots formation we need to use a more complex model than the overlap between different several single feature group models.

### The inclusion of distant features improves prediction power of the models

For cancer breakpoint hotspot modeling we used 100 kB coverage for all considered features as predictors. We decided to check whether model performance could be improved by the addition of different feature transformations to a current feature set (hereinafter referred to as local features). We created binary flags of feature presence, indicators of local/global coverage maxima and distant features (coverage aggregates on 1Mb window). We observed that a binary feature addition made no contribution to model quality while maximum indicators gave only slight performance increase (S12A Fig). On the contrary, distant features being added to local features improved ROC AUC by 0.03 on the average. However, the effect differs between cancer types being clearly positive for pancreatic, ovary, breast and liver cancer and slightly negative for blood. Precision and recall metrics are also improved with the inclusion of distant features (see S1 File). Based on these results we included distant features in the overall feature set (designated with the prefix "upper_").

Feature importance analysis from Boruta selection results (S7 Fig) has shown that all top non-B DNA features (G-quadruplexes, direct repeats, Z-DNA, short tandem repeats, mirror repeats), HDNase, and 5'UTR are presented in both scales (local and distant). For the remaining features (all five TFs and four genomic regions) only distant aggregates ranked in the top list. These observations indicate that the information about broader landscape of the genomic regions is useful for breakpoint hotspots prediction. Moreover, it is especially meaningful when the distant are combined with the local features, because it enables inside-model comparison of densities and identification of anomalies. Feature importance distribution for non-B DNA-based, TF-based, and genomic region-based breast cancer hotspots prediction model (Fig 4) shows that the model considers distant features as the most powerful factors while taking into account local features leads to further gain.

Additionally, we checked whether binary flags alone or these combined with the indicators of local/global maxima could help to predict cancer breakpoint hotspots (S12B Fig). The results showed that for models using only binary features the quality dropped by 0.13 ROC AUC on average (compared to local features), while the addition of indicators of maxima reduced the difference to 0.03 ROC AUC. This means that in itself the presence o a feature in a genomic area around a hotspot or the presence of a feature together with indicators of its maxima is less informative than the exact feature coverage in given genomic region.

### Non-predictable individual breakpoints and highly predictable breakpoint hotspots

Previous attempts to build models using full breakpoint distribution did not achieve good prediction performance (regression task, R2 of 10% for all cancers except for 18% for the breast cancer [9]). The modeling presented above was done for predicting breakpoint hotspots that correspond to genomic regions with recurrent mutations. Here we address the question

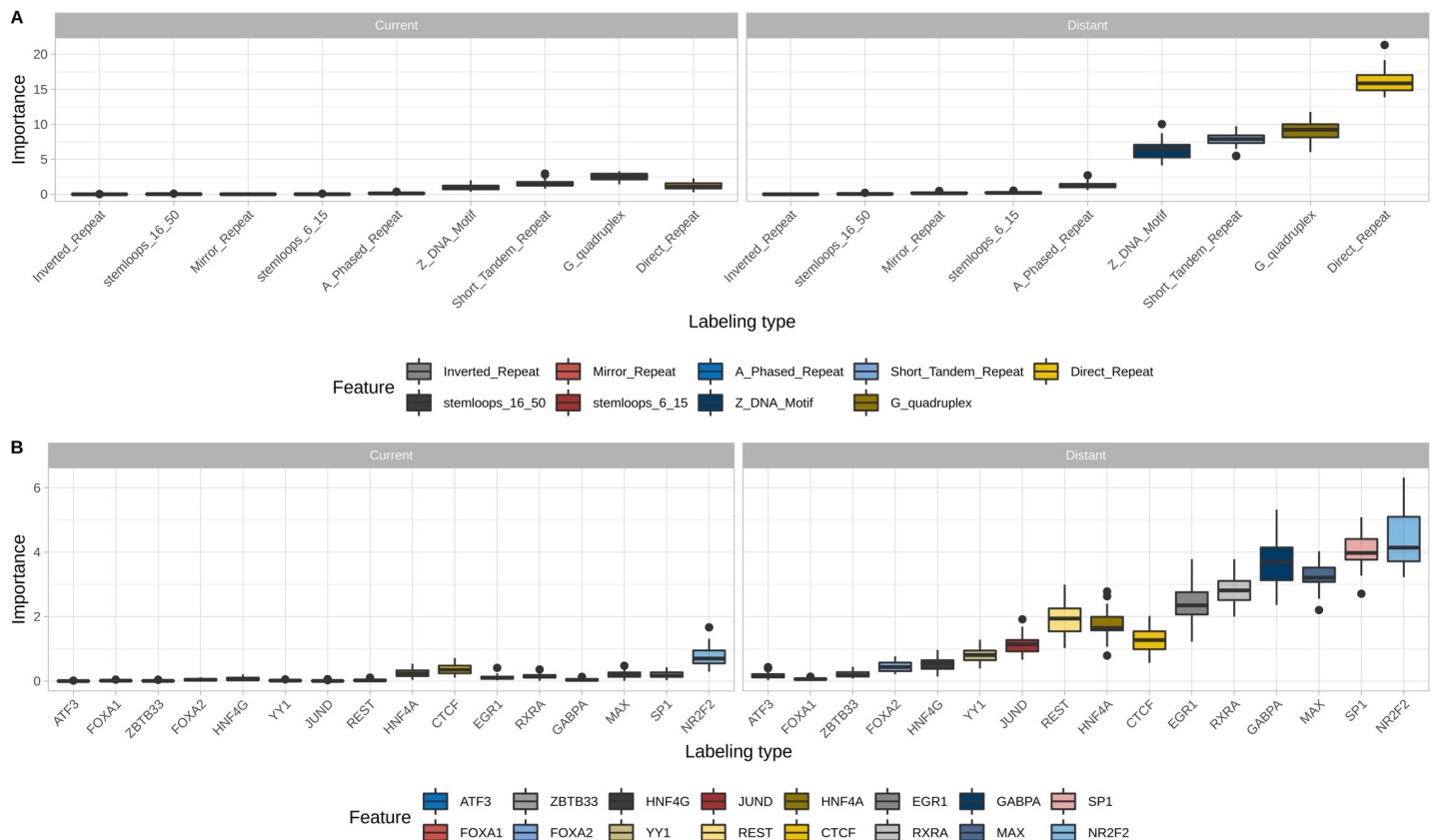

**Fig 4. Feature importance analysis inside non-B DNA, TF and genomic region groups.** A. Feature importance of models built on non-B DNA only for breast cancer (99% labeling type). B. Feature importance of models built on TFs only for breast cancer (99% labeling type). . Feature importance of models built on genomic regions only for breast cancer (99% labeling type).

whether individual breakpoints are just random noise or they also have general mutation determinants that can be captured by a machine learning approach. We performed a series of experiments for the comparison of models predicting breakpoint hotspots, individual breakpoints (S10 Table) and models that distinguish breakpoint hotspots from breakpoints (Fig 5 and S11 Table). The results show that for the majority of cancers the breakpoints are indeed unrecognizable from random locations (ROC AUC around 50% or slightly higher and the mean lift of recall ranging from 0 to 2.5), however for breast, ovary and prostate cancers even individual hotspots can be predicted with ROC AUC 65–75% (nevertheless the lift of recall is very low). In addition, models that can distinguish hotspots from breakpoints can be as good as models predicting hotspots, and reach 85% ROC AUC for breast cancer and more than 70% for brain, liver, pancreas, and prostate confirming that hotspots locations are completely different from individual breakpoints locations by considered features.

Besides it was established that for the majority of cancer types (all except bone, blood and brain) both ROC AUC and the median lift of recall increase with an increase in hotspots labelling threshold meaning that hotspots of higher breakpoints density differ more from other genomic regions than those of lower density (S13 Fig and S12 Table). As an example, decreasing the breakpoint density threshold for hotspot labeling by only 5% leads to 2 times lower mean lift of recall and 15% lower mean ROC AUC for all cancer types on average. Moreover, for considered low labeling thresholds ROC AUC exceeds 60% only for bone cancer while on average for all cancer types ROC AUC equals 54% with mean lift of recall of 1.6.

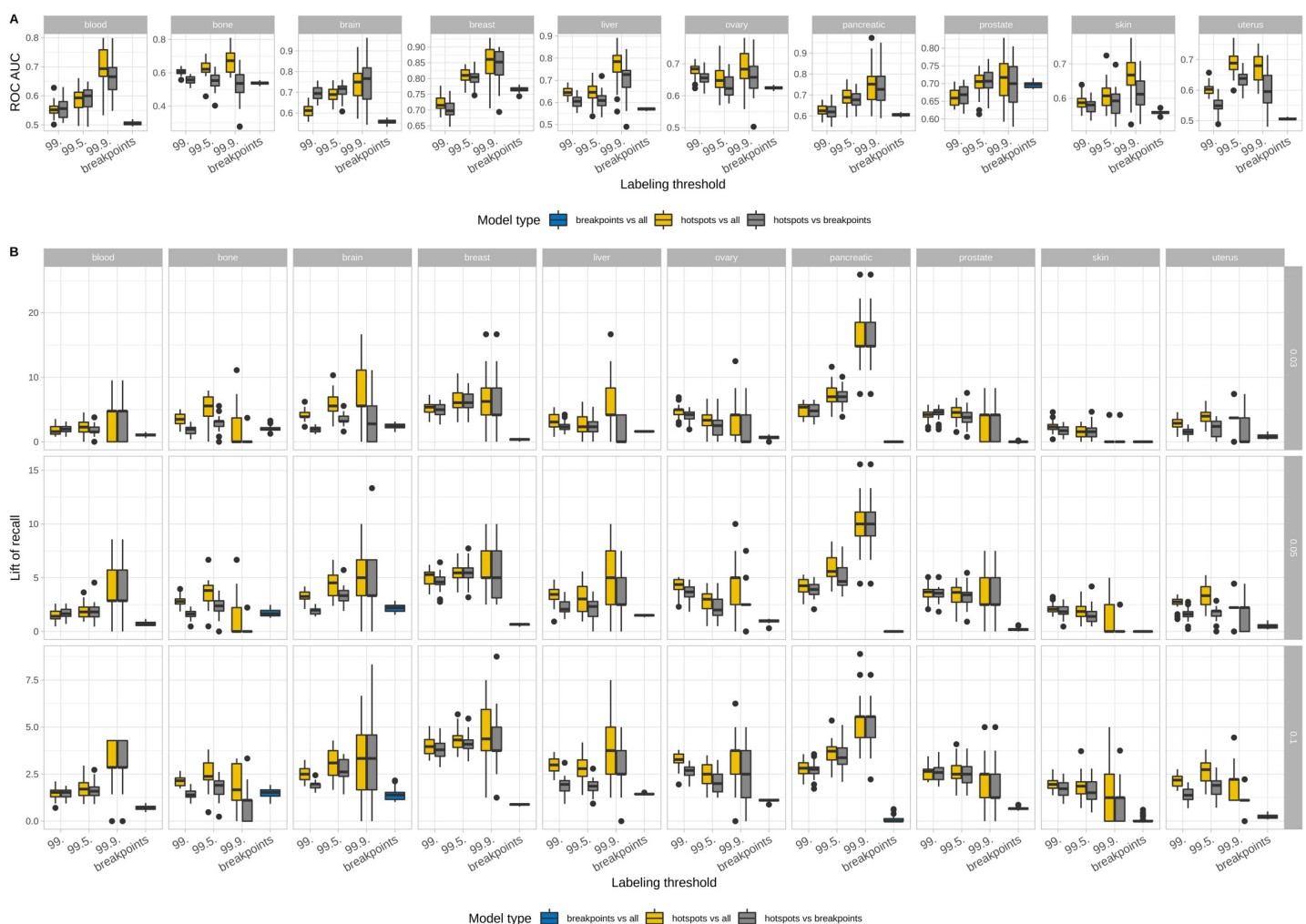

**Fig 5. Comparison of models predicting individual breakpoints and hotspots.** Distributions of model prediction power are presented for all cancer types using ROC AUC (upper panel) and lift of recall (lower panel). Blue colour signifies models distinguishing between breakpoints and random genomic positions ("breakpoints vs all"), yellow colour signifies models distinguishing between breakpoint hotspots and and random genomic positions, grey colour signifies models distinguishing between individual breakpoints and breakpoint hotspots. The results are presented for 0.03, 0.05 and 0.1 probability percentiles.

The results of these experiments are summarized in Table 1 (the median test ROC AUC) and Table 2 (the median test lift of recall for 0.03 probability percentile).

Randomness of breakpoint formation and breakpoint recurrence degree influence overall prediction model quality, and our results indicate that only breakpoints with high recurrence could be well explained by genomic and epigenomic features. Nevertheless, for some cancer types we achieved moderate quality of individual breakpoints prediction, which means that even on an individual level of breakpoints there exist structural mutation signatures underlying these processes, and it would be of interest to further research, distinguish and quantify the contribution of those structural mutation signatures in cancer mutagenesis.

## Comparison with the state-of the art approach

The latest comprehensive study that explored the contribution of different factors in cancer mutagenesis with machine learning models was the study of Georgakopoulos-Soare et al. [9], in which the authors investigated contribution of non-B DNA structures and histone

**Table 1. Median test ROC AUC for different models.**

| Cancer type | Breakpoints | 75. | 90. | 95. | Hotspots |
|---|---|---|---|---|---|
| Blood | 0,505 | 0,504 | 0,518 | 0,529 | 0,753 |
| Bone | 0,537 | | 0,675 | 0,715 | 0,644 |
| Brain | 0,559 | 0,590 | | 0,539 | 0,748 |
| Breast | 0,766 | 0,506 | 0,513 | 0,546 | 0,829 |
| Liver | 0,572 | | 0,529 | 0,564 | 0,710 |
| Ovary | 0,625 | 0,500 | 0,505 | 0,515 | 0,688 |
| Pancreas | 0,604 | 0,505 | 0,516 | 0,532 | 0,747 |
| Prostate | 0,698 | 0,500 | 0,502 | 0,509 | 0,664 |
| Skin | 0,559 | 0,506 | 0,508 | 0,530 | 0,569 |
| Uterus | 0,505 | | 0,519 | 0,569 | 0,701 |

modifications, either individually or in combination. As of note, Georgakopoulos-Soare et al. focused on somatic point mutations, and reported the results of the modeling for point mutations, with breakpoint analysis reported in Supplementary Materials. We compared our approach that uses more groups of factors and considers long-distant features to the data analyzed by Georgakopoulos-Soare et al. The results are presented in Fig 6. First we processed the raw breakpoints data (genomic coordinates) provided by Georgakopoulos-Soare et al. with our preprocessing pipeline and get breakpoints density for 500 kb length genome windows. We added all features used in our models (see Methods) and built a Random Forest regression following the model in [9] (prediction of breakpoint densities based on densities of different features) and using only non-B DNA and histone modification groups. We further built models using all of the features, including all long-distant features (Fig 6A). As it was done in [9], we used log2(1+x) for feature densities while we applied log2(mean(x)+x) transformations for the target variable. The model quality was assessed through a 10-fold cross-validation procedure. As a result, even on two feature groups–non-B DNA and histone modifications–the model achieved a better prediction power, partially because the size of the data sets is different since the data are constantly growing in ICGC, and partially because we used all available data on histone modifications and, additionally, the annotations by short stem-loop structures for non-B DNA group (see Methods).

Next we tested the prediction power of classification tasks that we used in the presented study–by selecting breakpoint hotspots based on the distribution thresholds (Fig 6B). Qualitatively we obtained the same results when the best model was the model that used all feature

**Table 2. Median test lift of recall for different models for 0.03 probability percentile.**

| Cancer type | Breakpoints | 75. | 90. | 95. | Hotspots |
|---|---|---|---|---|---|
| Blood | 1,055 | 1,070 | 1,587 | 1,235 | 4,762 |
| Bone | 1,921 | | 4,762 | 3,030 | 4,762 |
| Brain | 2,449 | 4,211 | | 2,564 | 5,556 |
| Breast | 0,366 | 0,498 | 1,254 | 1,444 | 6,061 |
| Liver | 1,599 | | 1,617 | 2,114 | 8,333 |
| Ovary | 0,674 | 0,152 | 0,691 | 0,836 | 4,981 |
| Pancreas | 0,000 | 0,891 | 1,726 | 2,096 | 18,519 |
| Prostate | 0,000 | 0,191 | 0,556 | 0,851 | 4,023 |
| Skin | 0,000 | 0,853 | 1,171 | 1,537 | 2,682 |
| Uterus | 0,734 | | 1,491 | 2,749 | 3,968 |

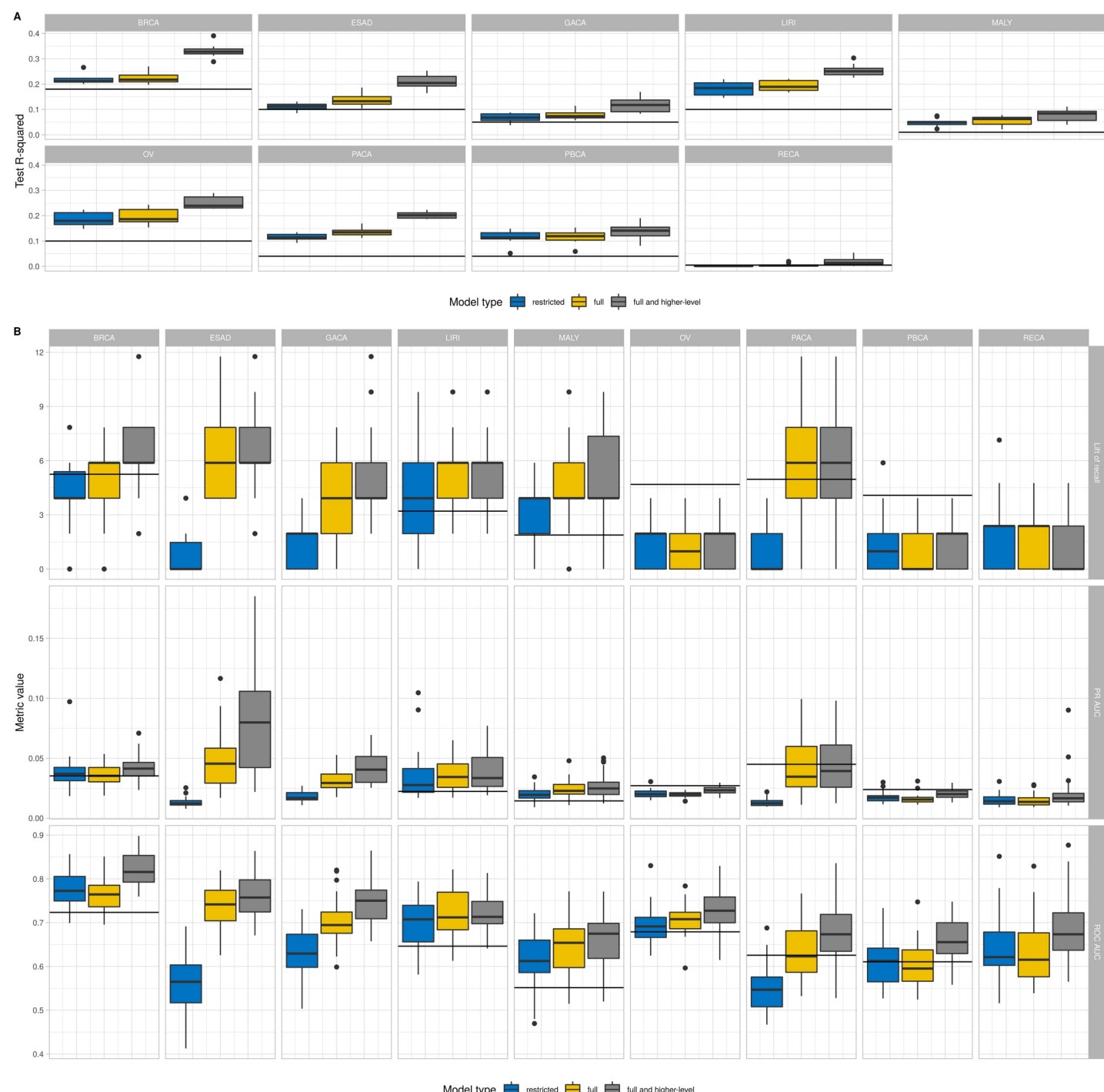

**Fig 6. Comparison with the state-of the art approach.** The state of the art is the model in Georgakopoulos-Soare et al. [9]. To enable comparison the designation of the types of cancer is given as in Georgakopoulos-Soare et al. The horizontal line mark the median performance of the model of Georgakopoulos-Soare et al. We built models using different groups of predictors: restricted (non-B and histone groups), full (all groups used in this study), and full and higher-level (all groups considering locations up to 1Mb) A. Comparison of regression models (prediction of breakpoints density). B. Comparison of classification models (prediction of 99% labeling type hotspots).

groups with additionally included distant features. The higher performance of the models trained in our study is explained by the increase in the data accumulated in ICGC since the publication of Georgakopoulos-Soare et al. [9].

## Discussion

Here we present a comprehensive analysis of cancer breakpoint hotspots with machine-learning approach using predictors from omics data. We have demonstrated that the best predictive power of machine-learning models is achieved when all groups of factors are taken into account. Stratification of group influence into predictive power of models points to the existence of different mutational mechanisms, or signatures, underlying the structural variation in cancer genomes. A detailed investigation of the contribution of different groups and individual features to breakpoint hotspot prediction led to some general trends that can be distinguished despite cancer heterogeneity. This heterogeneity, which is observed in mutational landscapes of cancer genomes, also shows in cancer-specific contribution of different groups and individual features.

As a common trend we observed two major groups of factors that contribute most to cancer breakpoint formation–non-B DNA structures and transcription factor binding sites. The role of non-B DNA structures in cancer mutagenesis was reported earlier in different studies [12,21], not only in formation of structural variants but also in that of point mutations [9]. The stratification by individual features highlighted G-quadruplexes and repeats (consistent with the previous reports [10–12,14]), but other factors must also be taken into account (Z-DNA and stem-loops).

Repeats are difficult to map and although structural variant calling pipelines take into account information on low mappability regions, we performed an additional analysis by filtering breakpoint data with low mappibility regions. We used 35-mer Duke Uniqueness track from UCSC genome browser (https://genome.ucsc.edu/cgi-bin/hgFileUi?db=hg19&g=wgEncodeMapability) and found that 8–13% of breakpoints fall into the low mappable regions–regions with less than 0.5 uniqueness score (S16 Table). Additionally, we conducted the entire analysis for the filtered data set (S17 Table) on the example of breast cancer. Boruta feature selection method confirmed non-B DNA feature group as the most important one with repeats remaining in the top10 list, following quadruplexes and preceding Z-DNA (S14 Fig). The latest studies confirm that CNVs are enriched in low mappability regions and repeats [22], and the fact that SVs often fall in repeat regions has by now been verified by long-read sequencing methods [23]. These findings suggest that in our study, due to the filtering procedures in SV calling pipelines, we have underestimated the proportion of breakpoints associated with repeats.

Transcription factor binding sites emerge as more important than histone modifications, underlying the causative role of transcription in breakpoint formation processes. The top 5 important TFs observed in more than 10% of all cancer genomes include SP1, RXRA, NR2F2, GABPA, and CTCF. All these factors are known to be associated with different cancers. SP1 is a general TF that is required for transcription of a large number of 'housekeeping genes', and it is over-expressed in many cancers [24–26]. For RXRA, from a group of vitamin D-related genes, associations were reported for leukemia [27], breast [28] and lung cancer [29]. Expression of NR2F2 was reported as positive for ovarian, while negative for prostate and breast cancers (Safe, Jin et al. 2014), though its role in cancerogenesis is not fully understood [25,26]. GABPA was associated with multiple cancers [30] and specifically with glioblastoma [31]. CTCF, a well-known regulator of 3D chromatin structure is associated with primary tumors (Achinger-Kawecka and Clark 2017; Aitken, Ibarra-Soria et al. 2018), and CTCF binding sites

are mutation hotspots in cancer [32]. The contribution of TF group captured with our machine learning approach reveals that cancer breakpoint hotspots are often located in the vicinity of known TF binding sites, and in this regard it is not surprising that the third group of factors from the top 3 comprise genomic positions, with 5'UTR and promoter regions being the leaders.

Histone modifications followed TFs and rated as top 3 only in four cancers–blood, brain, liver, breast. For blood the most influential factor was H3K36Me3 (gene marker) followed by H3K4Me3 (regulation of gene expression) and H3K9Me3 (heterochromatin marker). For brain the first most important was heterochromatin marker H3K9Me3 followed by H3K27me3. However, HM group is less prevalent and none of the important histone markers pass the threshold of being shared by 10% (300 out of 3000) datasets. Conversely, histone markers were shown to be the main contributors in predicting cancer point mutations [5,9]. This highlights the differences in cancer breakpoint and somatic point mutation mechanisms. Another epigenetic feature, methylation, is also a noticeable contributor in several cancers (the second most important for skin and breast and the third one–for brain).

The importance of genomic regions, especially of 5'UTR and coding exons is definitely correlated with TFs and histone modifications that mark active transcription. Coupled with non-B DNA structures, these results provide a hint towards a major mutagenic process when breakpoints are generated as a result of RNA polymerase stalled on quadruplexes or stem-loop structures formed by repeats. The distribution of breakpoints and breakpoint hotspots (see Figs 2 and 3 in [14]) have shown that the breakpoints are scattered across genomes with a probability of genomic region coverage. These are slightly underrepresented (48% vs 56%) in the gene bodies, although no significant enrichment/depletion is observed. However hotspots are underrepresented in gene bodies (only 10%), and in genes they are located mostly in introns. That is why the genomic region feature group did not rank as the top influential. GO enrichment analysis [33,34] of gene harboring hotspots revealed significant enrichment in the regulation of cell cycle transitions, the regulation of viral process, neurogenesis, phosphorylation and some other processes (see S13–S15 Tables). For molecular function, enrichment is found for many types of binding–enzyme binding, RNA binding, nucleic acid binding, metal ion binding, organic cyclic compound binding. Altogether these findings support the idea that DNA breaks mostly occur during active genomic processes including replication.

As to chromatin organization reflecting in HDNase densities and TAD distributions, these groups are less informative as compared to point mutation prediction. TADs influence is negligible and HDNase is not included in the top 3 list. CTCF, associated with 3D chromatin structure, is the 5[th] TF by the order of importance and overall presence in the analyzed genomes.

Another important conclusion to draw from our analysis is the importance of considering long-distance (up 1 Mb) action of various factors. The initial modeling was done using 100 kB regions, and we showed that adding additional long-range to short-range acting features improved the predictive power of models reflecting complex organization of genetic information. Moreover, we demonstrated that for major contributing feature groups (non-B DNA and TF) distant features were more important than local ones almost for all cancer types. This fact could be explained on the one hand by the presence of active regulatory elements such as enhancers that are located at a long distance from a gene in DNA sequence, but these elements are found in proximity to the gene in chromatin structure. It is also indicative of transcriptionally active large-scale regions, and it will be a subject for further study to locate and analyze these regions from RNA-seq data from cancer patients.

Though we have managed to improve models by aggregating information from omics data, the precision and recall metrics remained low with an average 2% (up to 5%) precision and an

average 11% (up to 50%) for recall (S7 and S8 Tables), resulting in very low values for PR AUC. This means that the model better predicts the second class, or in other words it better recognizes regions without breakpoints than breakpoint hotspots. Low precision corresponds to a high number of false positives–which means that the model calls a region being a breakpoint hotspot region when it is not. This could be partially due to the fact that this region has all the properties (genomic and epigenomic), which are characteristic for hotspots, and they could be potential breakpoint regions that have not yet been documented. Low classification metrics reflect the fact that with our set of preselected features we still did not include enough predictors of breakpoint mutagenesis. However we observe the correlation between the number of data available and an increase in model predictive power as it is with the breast cancer that has around 20,000 breakpoints compare to 2,000 for blood. Also we were limited by the available epigenomic and transcriptomic data sets, which will also grow in size and will add to model performance in the future. Building a machine-learning model to predict cancer breakpoints with high predictive power still remains a challenge.

Difficulties in building machine learning models to predict cancer breakpoints raises the question about degree of randomness in breakpoint landscape. We tried to resolve this by comparing models predicting individual breakpoints and breakpoint hotspots. Hotspots are always better recognizable than single breakpoints and moreover we can distinguish hotspots from single breakpoints with a high degree of confidence. Nevertheless, the elected machine learning approach showed that for some cancer types (breast and prostate) individual breakpoints can be predicted even with 0.70 ROC AUC, suggesting that there exist determinants of the structural mutagenesis process. It will be of interest to further investigate which cancer genomes have a higher degree of randomness in terms of breakpoint prediction, and what are the major mutagenic processes for those breakpoints that can be predicted.

Despite the general trends of commonly shared important groups of factors, at the level of individual cancer types we observe various degrees of contribution of different factorsm that emphasize the heterogeneous nature of cancers. The model predictive power of each group and of each feature inside the groups can be converted to cancer-specific breakpoint mutation signatures.

It is important to note that in this study we analyze non-cancerous distributions of features and cancerous breakpoints mapped to these distributions. In doing so we try to answer the questions–which regions of the so-called normal genomes are more susceptible to breakpoint formation during cancerogenesis. Cancerous genomes undergo changes in epigenetics, and it would be of interest to investigate how these changes in epigenetic landscapes are associated with breakpoint formation. Big data analysis retrieves the most important signals and reveals major trends in the data (as found for the influence of non-B structures on transcriptional processes), however in the future when more data on single cancerous genomes become available, more information should become retrievable.

A recent report from PCAWG Consortium identifies 16 signatures of structural variation [35], which include size distribution of different classes of structural variants such as deletions, tandem duplications, copy number gains as well as association of cycles of templated insertions with the gene TERT activation. The next step in understanding is to reveal associations between signatures and mutagenic processes. Our results demonstrated that formation of non-B DNA structures most likely coupled with transcription processes are the major determinants in cancer breakpoint formation, and epigenetics has much smaller predictive power. Varying contribution of each feature group is observed for major cancer types. Breakpoint hotspots in brain can be predicted by the distribution of non-B DNA structures, those in liver–by transcription factor binding sites, those in the blood–by non-B DNA structures and promoter regions. It would be of importance for future research to identify structural mutational

signatures and decompose contributions of each feature group to mutation signatures in individual patients and cancer types.

## Methods

### The data

To form a target variable we used the data for 628,126 cancer genome breakpoints from 10 cancer types (release 28 from ICGC [36]). Compared to previous analysis [14] performed on an earlier data release, the sample was enriched with more than 165,000 new breakpoints, which became available mostly for prostate cancer (more than 128,000 new breakpoints) (S1 Table). The breakpoint data were pre-processed prior to the analysis. First, the data were filtered so that breakpoints with a high position inaccuracy (a start/end location interval is wider than 10 bases) were removed from the sample (S2 Table). Then the genome was split into non-overlapping windows 100 kb wide, and further analysis was performed at the level of genome window.

To control for genome instability introduced by natural topological constrains some windows were removed. Specifically, we excluded the windows intersecting with centromeres and telomeres, the first and the last window of each chromosome. Additionally, we took into consideration the DAC blacklist (https://www.encodeproject.org/annotations/ENCSR636HFF/) comprising a set of genome regions recognized as low mappable or anomalous and recommended to be removed from genome studies.

Concerning the blacklisted regions, it is worthy of notice that the percent of breakpoints in these regions is very small. So it is minimal for liver cancer with 0.09% of breakpoints located in the regions, maximal for brain cancer (6,8%) and is in the range of 0.4% - 1.5% for other cancer types. Finally, Y and MT chromosomes were not included into the study as they contain relatively small number of breakpoints. Finally, we selected 29288 100-kb windows.

To test whether low mappability regions affect the set of top features determined with machine learning algorithms we filtered breakpoints falling into Duke 35-mer track with mappability less than 0.5 (https://genome.ucsc.edu/cgi-bin/hgFileUi?db=hg19&g=wgEncodeMapability).

For each window located in a specific chromosome, breakpoint density was calculated as a number of breakpoints overlapping with the window divided by the total number of breakpoints in the chromosome separately for each cancer type. Given breakpoint density we highlighted the hotspots–the regions of the genome significantly enriched with breakpoints. The window was designated as a hotspot if its breakpoint density is greater than the specified density distribution percentile (99%, 99.5%, 99.9%) for corresponding cancer type. These three thresholds are referred to as labeling types (labeling thresholds). The total number of breakpoints sorted by cancer type and labelling type is given in S3 Table. Additionally, binary labels were assigned to genome windows indicating the presence/absence of breakpoints.

Genomic features data were downloaded from several projects such as the Encode [18], DNA Punctuation (www.dnapunctuation.org)[17], Non-B DB projects [16], UCSC Genome Browser, and from materials of research papers [19]. The source data for secondary structure annotations for all types except for stem-loops were collected from Non-B DB, while the latter were provided by the DNA punctuation project. Then we used genomic regions markup from UCSC Genome Browser. Topologically associating domain (TAD) data were published in [19] (S3 Table "TAD boundary annotations"). The epigenetic features data were downloaded from the Encode project. We selected experiments with released status, a number of biological replicates greater than one, with no treatment applied. We used TF ChIP-seq for transcription

factors,—DNase-seq for DNA accessibility,—Histone ChIP-seq for histone modifications, and RRBS for DNA methylation–.

Feature data were aggregated in chromosome windows by calculating the coverage as the summary length of all regions in a window occupied by a feature (avoiding double account of overlapping regions) divided by the window length, so that the coverage reflects how many bases in a window are covered by the feature. Feature densities were calculated as a percent of feature coverage of a genomic region of 100kB (local features) or 1 Mb (distant features). Based on these densities we obtained binary flags (1, if coverage of the genomic characteristic is greater than 0, and 0 otherwise) and local/global maximum indicators (a binary indicator of the feature value exceeding 90%/ 95%/ 99% percentiles of the feature distribution; a binary indicator of the feature value exceeding values of all 1/5/10 preceding and following adjacent genomic regions; the relative difference between the feature value and the maximum of 10 preceding and following neighbouring regions).

A list of groups and features available for each cancer type is presented in S5 and S6 Tables. The total analysis was performed for 30 datasets (10 cancer types with 3 different target labeling types) comprising hotspot binary labels and genomic feature coverage. The total number of features for each cancer type is given in S6 Table.

## Modeling

We used Random Forest [37] model for the machine-learning approach, with hyperparameters (maximal number of terminal nodes, number of trees, minimal number of observations in a terminal node, number of sampled variables for a tree) selected to be optimal when averaging ROC AUC statistics across all cancer types using data for 99% labeling type.

To attain reliable model performance estimates we used a 30-times repeated and stratified train-test split with 30% of data in test sample, with the splits stratified by chromosome and the inside position The quality of models was measured on each of 30 test sets with ROC AUC and lift of recall/lift of precision metrics. These metrics show the number of times recall/precision of the model is greater than a random choice using specific probability percentile as a threshold where each probability percentile is given by the percentile of the model output probability distribution. In particular, the lift of recall for a specific probability threshold (given by a percentile of a probability distribution) is calculated as the recall divided by this percentile of the probability distribution. For example, since it is supposed that in case of a random choice recall approximates the probability percentile (labeling n% of examples gives n% recall), by dividing the model recall by the probability percentile we get the number of times a model is better compared to a random choice.

After training and evaluating hotspot prediction models based on all features (S15 Fig and S18 Table) we performed Boruta feature selection [38] to choose the most important features for hotspot prediction task for each type of cancer. Boruta feature selection is a method specifically designed for Random Forest which for various random initializations iteratively checks whether a considered feature is more important than all of the random ("shadow") features and updates the feature set for the next iteration by removal of unimportant features. Based on this approach we selected top features for each type of cancer (based on results for 99% labeling type) and evaluated models on these feature sets (S16A Fig and S19 and S20 Tables). It was established that for three cancer types (pancreatic, prostate and breast cancer) the quality of models dropped in comparison with the quality of models on all features. To eliminate the difference we performed forward feature selection and found 1, 1 and 2 features respectively which addition to feature sets lead to comparable quality (S16B Fig and S21 Table). These final features sets for each type of cancer are referred to as "all" in Fig 3.

Feature group importance analysis was done as follows. We built models separately on features of each feature group for each cancer type. To obtain feature group performance rankings we fixed the maximum achievable mean lift of recall at 0.03 probability percentile for each labeling type and scaled the values of the mean lift of recall for all feature groups to that fixed value and then averaged for the cancer type for 99% and 99.5% labeling types (Fig 1).

For the analysis of the degree of randomness of cancer breakpoints we performed a set of experiments. First, we built models to predict hotspots with increasing hotspot labeling thresholds to 75%/90%/95% of breakpoint density distributions. Secondly, we estimated the quality of models discriminating hotspots from breakpoints and models discriminating breakpoints from the remaining genomic areas to compare it to performance of hotspot prediction models. In both cases for modeling we used all available features. At the end we compared all performance metrics and identified cancer types for which random nature of breakpoints formation with predictable recurrent locations is the best visible (S17 Fig).

## Supporting information

**S1 Fig. Features Spearman correlation.**
(TIFF)

**S2 Fig. Distributions of mean lift of recall and ROC AUC.** A. Distribution of mean lift of recall for 0.03 probability percentile by feature group and labelling type over all cancer types. B. Distribution of mean ROC AUC by feature group and labelling type over all cancer types.
(TIFF)

**S3 Fig. Distribution of feature importance for non-B DNA models.** Distribution of feature importance (from Random Forest model) for models based on non-B DNA features. The order of features is preserved in each panel of plot.
(TIFF)

**S4 Fig. Non-B DNA features coverage with cancer hotspots labels.** A. Non-B DNA features (quadruplex, Z-DNA) coverage with cancer hotspots labels for a region of 13 chromosome. B. Non-B DNA features (Direct Repeat, Mirror Repeat, Short Tandem Repeat, stem-loops) coverage with cancer hotspots labels for a region of 13 chromosome.
(TIFF)

**S5 Fig. Distribution of feature importance for TF models.** Distribution of feature importance (from Random Forest model) for models based on TF features. The order of features is preserved in each panel of plot.
(TIFF)

**S6 Fig. Boruta feature importance analysis.** Number of executions in which a feature was recognized as important by Boruta feature selection in hotspots prediction models.
(TIFF)

**S7 Fig. Ranking of Boruta important features by the number of datasets.** The number of datasets for which feature was considered as important in Boruta feature selection procedure for all cancer types. Only features with the total number of datasets no less than 300 are included.
(TIFF)

**S8 Fig. The number of datasets by cancer type for top 8 features by total number of datasets in Boruta feature selection.**
(TIFF)

**S9 Fig. Model performance based on one feature group and on Boruta selected features.** Distribution of lift of recall for 0.03 probability percentile, PR AUC and ROC AUC for each of one group-based models and models in Boruta selected features (for 99.5% labeling type). (TIFF)

**S10 Fig. McFadden's R-squared for hotspots prediction models.** Median values of McFadden's R-squared for modelling hotspots (99% and 99.5% labelling type) with logistic regression. (TIFF)

**S11 Fig. Dependency between different feature group based models predictions.** A. Mean Spearman correlation for predicted probabilities of genome window being a hotspot for different feature group based models for 99% labeling type. B. Distribution of precision at threshold corresponding to selection of 5% of genome windows with the highest probability for 99% labeling type. At x-axis the number of intersected feature group based models predictions is depicted (label "1" corresponds to best single feature group based model predictions while label "2" denotes the genome windows where exactly two feature group based models predictions intersected). Annotation—number of selected genome windows marked as hotspots. (TIFF)

**S12 Fig. Comparison of models using local, distant and binary features.** A. Distribution of difference between test ROC AUC of model on extended feature sets and on local features. B. Distribution of difference between test ROC AUC of model on binary features / indicators of maximums and on local features. (TIFF)

**S13 Fig. Comparison of different hotspot selection criteria.** Distribution of test ROC AUC and lift of recall for 0.03, 0.05 and 0.1 probability percentiles for hotspots prediction models with high and low hotspots labeling thresholds for each type of cancer. (TIFF)

**S14 Fig. Ranking of Boruta important features for breast cancer dataset with breakpoints in low-mappable regions removed.** The number of datasets for which feature was considered as important in Boruta feature selection procedure for breast cancer (99% labelling type). (TIFF)

**S15 Fig. Distribution of performance metrics of models based on all features.** Distribution of performance metrics for hotspots prediction models based on all features by cancer type and hotspots labelling type. (TIFF)

**S16 Fig. Model comparison based on various set of features.** A. Comparison of mean test lift of recall and mean ROC AUC for hotspots prediction models trained on all features and on reduced sets of features (the best feature sets produced by Boruta feature selection) for 0.03 probability percentile threshold. B. Comparison of mean test lift of recall and mean ROC AUC for hotspots prediction models trained on all features and on reduced sets of features (the best feature sets produced by Boruta feature selection extended with several important features) for 0.03 probability percentile threshold. (TIFF)

**S17 Fig. Comparison of "hotspots vs all" and "breakpoints vs all" prediction models.** A. Ratio of lower bound of confidence interval for mean lift of recall for "hotspots vs all" prediction model to median lift of recall for "breakpoints vs all" prediction model at 0.03, 0.05 and

0.1 probability percentile. Gray colour corresponds to infinity value meaning that is equal to zero. B. Ratio of lower bound of confidence interval for mean lift of recall for "hotspots vs all" prediction model to lower bound of confidence interval for mean lift of recall for "hotspots vs breakpoints" prediction model at 0.03, 0.05 and 0.1 probability percentile. For visualization purposes two negative outliers (-27.82 and -1.05) were set to 0.5 and 4 positive outliers (from 3,39 to 8,12) were set to 3.
(TIFF)

**S1 Table. Summary of cancer-related studies aimed at point mutations or breakpoints analysis.**
(XLSX)

**S2 Table. Summary statistics for breakpoint dataset by cancer type.**
(XLSX)

**S3 Table. Summary statistics for preprocessed breakpoint dataset by cancer type.**
(XLSX)

**S4 Table. Number of hotspot windows by cancer type and labeling type.**
(XLSX)

**S5 Table. Number of genomic features by cancer type.**
(XLSX)

**S6 Table. List of genomic features and its availability for different cancer types.**
(XLSX)

**S7 Table. Performance metrics for feature group models.** Performance metrics for hotspot prediction models for each labeling and cancer type by each feature group separately.
(XLSX)

**S8 Table. Performance metrics for best feature set models.** Performance metrics for hotspot prediction models for each labeling and cancer type using best selected features from full feature set for each type of cancer.
(XLSX)

**S9 Table. Median values of McFadden R-squared for logistic regression based models for 99% and 99.5% labelling types hotspots prediction by feature group.**
(XLSX)

**S10 Table. Performance metrics for individual breakpoint prediction models by cancer type.**
(XLSX)

**S11 Table. Performance metrics for hotspots vs breakpoints prediction models for each type of cancer.**
(XLSX)

**S12 Table. Performance metrics for hotspot prediction models for each type of cancer with low hotspots labeling thresholds.**
(XLSX)

**S13 Table. GO analysis of biological processes of genes harboring breakpoint hotspots.**
(XLSX)

**S14 Table. GO analysis of molecular function of genes harboring breakpoint hotspots.**
(XLSX)

**S15 Table. GO cellular component analysis of genes harboring breakpoint hotspots.**
(XLSX)

**S16 Table. Number of breakpoints in low mappability regions by cancer type and uniqueness score.**
(XLSX)

**S17 Table. Breast cancer breakpoints dataset filtered for low mappability regions.**
(XLSX)

**S18 Table. Performance metrics for hotspots prediction models for the best labeling type (by mean lift of recall) for each type of cancer for full feature set.**
(XLSX)

**S19 Table. Feature selection results.**
(XLSX)

**S20 Table. Performance metrics for Boruta-selected feature models.** Performance metrics for hotspot prediction models for the best labeling type for each type of cancer using only Boruta-selected features.
(XLSX)

**S21 Table. Performance metrics for final feature set models.** Performance metrics for hotspots prediction models for the best labeling type for each type of cancer using final feature set.
(XLSX)

**S1 File. Supplementary Methods.**
(PDF)

## Acknowledgments

We thank Serena Nik-Zainal, Ilias Georgakopoulos-Soares and Martin Hemberg for sharing data from the study [9]. Support from the Basic Research Program of the National Research University Higher School of Economics is gratefully acknowledged.

## Author Contributions

**Conceptualization:** Maria Poptsova.

**Data curation:** Kseniia Cheloshkina.

**Formal analysis:** Kseniia Cheloshkina, Maria Poptsova.

**Funding acquisition:** Maria Poptsova.

**Investigation:** Kseniia Cheloshkina, Maria Poptsova.

**Methodology:** Kseniia Cheloshkina, Maria Poptsova.

**Software:** Kseniia Cheloshkina.

**Supervision:** Maria Poptsova.

**Validation:** Kseniia Cheloshkina.

**Visualization:** Kseniia Cheloshkina.

**Writing – original draft:** Kseniia Cheloshkina, Maria Poptsova.

**Writing – review & editing:** Kseniia Cheloshkina, Maria Poptsova.

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
