## [Decision Letter · Decision Letter 0]

13 Jul 2020

Dear Dr. Poptsova,

Thank you very much for submitting your manuscript "Comprehensive analysis of cancer breakpoints reveals signatures of genetic and epigenetic contribution to cancer genome rearrangements" for consideration at PLOS Computational Biology.

As with all papers reviewed by the journal, your manuscript was reviewed by members of the editorial board and by two independent reviewers. In light of the reviews (below this email), we would like to invite the resubmission of a significantly-revised version that takes into account the reviewers' comments.

We cannot make any decision about publication until we have seen the revised manuscript and your response to the reviewers' comments. Your revised manuscript is also likely to be sent to reviewers for further evaluation.

Sincerely,

Anna R Panchenko

Associate Editor

PLOS Computational Biology

Florian Markowetz

Deputy Editor

PLOS Computational Biology

Reviewer's Responses to Questions

**Comments to the Authors:**

Reviewer #1: Manuscript Number: PCOMPBIOL-D-20-00990

Title: Comprehensive analysis of cancer breakpoints reveals signatures of genetic and epigenetic contribution to cancer genome rearrangements

Authors: Cheloshkina and Poptsova

The authors established a predictive model by using a machine learning approach from a set of cancer breakpoint data and other genomic DNA feature datasets, such as non-B DNA, transcription factor binding sites, and epigenetic modifications. The model was evaluated by its power to predict genomic breakpoints in cancer and was used to estimate the contributions of these factors in chromosome breakage in cancer. The authors report here that although there are obvious cancer type-specific characteristics, non-B DNA and TF binding sites are the most powerful factors in predicting recurrent breakpoint hotpots in most cancer types.

This is a very timely study, aimed to address a very important question in the field: what are the reasons for chromosomal breakage hotspots in cancer? However, the impact of this study would be increased by addressing the following comments/concerns:

1. The appearance of mutations (breakpoints) in cancer genomes seems to be biased: some DNA breakage events might be lethal and therefore are eliminated from genomes quickly; some are associated with cancer development and therefore are enriched in cancer datasets. Although the authors mentioned “genomic positions (whole genes, exons, introns…)” at the beginning of the manuscript, and provided some supplementary data, this important feature was not adequately discussed. Did the authors detect any differences in genes vs. intergenic regions? Functions and importance of impacted genes? Transcribed regions vs. non-transcribed regions? Translated regions vs. untranslated regions? All of these issues should be addressed by the authors.

2. The genomic features included in the study, i.e., non-B DNA, transcription factor binding sites, chromatin structures, and epigenetic modifications are somewhat associated with each other. For example, transcription occurs more frequently in open chromatin regions, which also have unique epigenetic modifications. And non-B DNA structures can affect all of these features. When evaluating each individual feature for its contribution to DNA breakage hotspots in cancer, how did the authors distinguish the impact of each feature? This needs to be addressed.

3. Maybe I have missed this information, but it needs to be clearly stated by the authors: were the data on transcription factor binding sites, chromatin organizations, and epigenetic modifications also from cancer samples? If they were from healthy individuals, what was done to establish the connection to cancer? The authors should clearly address these points.

4. Are there any “overlapped” regions identified by different features? Are these multiple feature regions more prone to breakage? The authors should address these questions.

5. Line 202: why do some features exhibit “distant aggregation only”? For example, TF binding sites are in general pretty short, why do they affect only distal regions but do not affect nearby regions? Is it possible that there are indirect associations? What are the consequences of TF binding, e.g., activation of transcription? The authors should discuss these points.

6. Can the authors provide information about the TF occupation at these sites in cancer cells?

7. When larger regions were included for long distance action, the “predictive power” was improved. Does that mean that this model could predict and identify more reported DNA breakage hotspots in cancer samples? How specific is the prediction? Did it also increase the rate of miss-hitting?

8. The fonts in the figures, particularly in Fig. 1, are too small to read.

Reviewer #2: The authors describe a comprehensive analysis of cancer breakpoints and their correlates with genomic and epigenetic features. The paper describes correlation levels of these features across cancer types, showing differences and similarities between cancer types. It is however not clear how new these insights are compared to previous models/analysis (ref 9-14 in the introduction). Perhaps the most interesting insight from the paper is the potential difference between breakpoint hotspots and singletons. However, this aspect is not analysed or described in-depth, and could have been a greater focus of the paper. Furthermore, I have multiple concerns regarding the use of test sets, accuracy metrics for imbalanced data, and benchmarking with other published models/methods.

## Not clear if test set is independent

It is not clear if the test sets are completely independent of the training sets. The authors should avoid information leakage between training and test sets. Furthermore, for reproducibly and subsequent model benchmarking/comparison by other groups, the authors should keep a separate withheld test set for reporting of AUC/AUPRC measures. The accuracy (AUPRC/AUC) in figures should be mainly derived from this test set. All data (breakpoints, cancer types, genomic coordinates) for this test set should be included in the supplement.

## Accuracy metrics and issues of class imbalance

The authors use AUC for assessing model accuracy. However, its not clear how the negative set was defined and how imbalanced the positive and negative sets are. This should be described in the methods. In their scenario, true negatives are likely arbitrary and greatly dominate the positive class. A more suitable accuracy metric in this setting is the area under the precision-recall curve (AUPRC). The authors should report AUPRC (in addition to, or instead of AUC) numbers throughout text and figures.

## R-squared not reported

To get a sense of how much of the variance these top-features capture, alone and in combination, the authors should also report measures of R-squared.

## Lack of clearly defined negative set

It is not clear how the negative set (regions without breakpoints) was defined for model training and testing.

## No comparison with previous studies/methods

The authors cite several previous studies analyzing associations between cancer breakpoints and epigenetic features. The authors should compare their models performance with relevant existing approaches/models/methods (e.g. references 9-14 in the paper).

## Concern that non-B DNA features are enriched for repeat regions

It is concerning that the non-B DNA feature group is largely composed of regions with repetitive sequences (e.g. Figure 2). This may cause mapping/alignment errors and affect breakpoint calling, i.e. a higher false-positive rate in repeat regions. The authors should consider the align/map-ability of these regions, and evaluate the impact/bias this has on this feature group. The authors should remove/filter all regions with low mappability from the analysis.

#### Minor comments

* Missing figure caption/text for Figure 2.

* P. 7 “When analyzing group feature importance for all cancers combined, the non-B DNA structures are on the first place followed by transcription factors and genomic regions (2A and 2B Fig)”. It is not clear from figure 2 that anything is in 1st or 2nd place. This should be quantified and explained better. Also, Figure 2 caption text is missing.

* P. 8 “”Independent from the methods – group or Boruta feature importance – two best performing feature groups were always non-B DNA and TFs.” How is this conclusion reached? This is also a key claim in the abstract. The previous paragraph refers to many different supplementary figures, but the authors do not provide specific and direct data (e.g. a statistical test) in a main figure to support this claim.

* Figure 4: It is difficult to evaluate the difference between local and distant features in these plots since the importance axis have different scales. Its also not clear if local and distant features were evaluated alone or in combination. The authors should run the importance analysis using _all_ features (both local and distant) and plot them both local/distant features in the same plot.

* P. 11: “Previous attempts to build models using full breakpoints distribution did not achieve good prediction performance.” Which attempts were these? Please explain.

* Figure 5: “ The results are presented for 0.03, 0.05 and 0.1 probability percentiles.” It is not clear what the x-axis in figure 5 denotes.

**Have all data underlying the figures and results presented in the manuscript been provided?**

Reviewer #1: Yes

Reviewer #2: **No: **For reproducibly and subsequent model benchmarking/comparison by other groups, the authors should keep a separate withheld test set for reporting of AUC/AUPRC measures.

PLOS authors have the option to publish the peer review history of their article (what does this mean?). If published, this will include your full peer review and any attached files.

Reviewer #1: No

Reviewer #2: No
---

## [Decision Letter · Decision Letter 1]

1 Oct 2020

Dear Dr. Poptsova,

Thank you very much for submitting your manuscript "Comprehensive analysis of cancer breakpoints reveals signatures of genetic and epigenetic contribution to cancer genome rearrangements" for consideration at PLOS Computational Biology.

In light of the reviews (below this email), we would like to invite the resubmission of a significantly-revised version that takes into account the reviewers' comments. The reviewers were not satisfied with your first round of revisions and I invite you to address the reviewer's points about the statistical significance of your results and other issues.

We cannot make any decision about publication until we have seen the revised manuscript and your response to the reviewers' comments. Your revised manuscript is also likely to be sent to reviewers for further evaluation.

Sincerely,

Anna R Panchenko

Associate Editor

PLOS Computational Biology

Florian Markowetz

Deputy Editor

PLOS Computational Biology

Reviewer's Responses to Questions

**Comments to the Authors:**

Reviewer #2: >> Reviewer: “The authors should report AUPRC (in addition to, or instead of AUC) numbers throughout text and figures.”

>> Authors: “We agree with the reviewer that AUCPR is often used in cases of imbalanced classes. In our case its use is inappropriate since AUCPR depends on the proportion of positive examples in a dataset and that makes it impossible to compare datasets with different class balance”

The authors attempt to discuss their way through this point. To avoid misleading the readers, they MUST report AUPRC for one of these splits, for example the middle split (1% positives). The authors can use the other reporting metrics when they compare across splits containing different positive proportions.

>> Reviewer: “To get a sense of how much of the variance these top-features capture, alone and in combination, the authors should also report measures of R-squared.”

>> Authors: “R-squared is not reported because of model specification: we predict binary labels of breakpoint/breakpoints hotspot presence in a genomic window, we can only measure binary classification metrics, not the regression one.”

The authors are not providing any help here. As a reader, I have no sense of how well these individual covariates explain the variance in the data. Do they explain 0.01% or 50% better than a naive/null model? Lift of recall, feature importance, and AUC (see above) are not intuitive metrics. There are interpretable metrics for binary data, such as McFadden’s R2 (“https://en.wikipedia.org/wiki/Logistic_regression#Evaluating_goodness_of_fit”). The authors should plot McFadden’s R2 for these top features.

>> Reviewer: “The authors should compare their models performance with relevant existing approaches/models/methods (e.g. references 9-14 in the paper).”

>> Author: “We summarized all the reviewed material in one table (new Supplementary Table S1) with benchmark of considered approaches for modeling and analysis of point and breakpoint mutations. Other researchers, who applied machine learning methods, aimed at prediction of breakpoints density and hence performed regression task which is incomparable with classification task we tried to solve. Our previous research … is the only classification models for comparison.”

Table S1 is a simple summary of existing methods with accuracy statistics copied from the respective papers. There is NO direct comparison/benchmark with the authors method on the same datasets. Firstly, while previous studies may have used regression to predict (continuous) breakpoint density, the authors have instead used (ad-hoc) discretisation of the density (using various cutoffs), and they therefore claim methods/results cannot be compared. This is NOT the case. The authors can run existing (regression-based) methods, set a cutoff for the prediction score to obtain positive and negative classes, and then compare with their own model. It is important that the authors compare their method to existing approaches so the reader can gauge the improvement above current state of the art.

>> Reviewer: “It is concerning that the non-B DNA feature group is largely composed of regions with repetitive sequences (e.g. Figure 2). This may cause mapping/alignment errors and affect breakpoint calling, i.e. a higher false- positive rate in repeat regions.”

>> Authors: “More detailed description was in Supplementary Methods, which we moved now to the main Methods… Additionally we took into consideration the DAC blacklist (https://www.encodeproject.org/annotations/ENCSR636HFF/) comprising a set of genome regions recognized as low mappable or anomalous and recommended to be removed from genome studies.“

The authors did no extra work to convince me this is not an issue. The encode blacklist is a very small and conservative list of regions. The authors could have analyzed/filtered the data with the mappability tracks from UCSC genome browser (e.g. “https://genome.ucsc.edu/cgi-bin/hgFileUi?db=hg19&g=wgEncodeMapability”), and showed me this had no effect on their conclusions about non B-DNA features. Their decision not do this makes we worry about the robustness of their conclusions.

**Have all data underlying the figures and results presented in the manuscript been provided?**

Reviewer #2: Yes

PLOS authors have the option to publish the peer review history of their article (what does this mean?). If published, this will include your full peer review and any attached files.

Reviewer #2: No
---

## [Decision Letter · Decision Letter 2]

16 Dec 2020

Dear Dr. Poptsova,

Thank you very much for submitting your manuscript "Comprehensive analysis of cancer breakpoints reveals signatures of genetic and epigenetic contribution to cancer genome rearrangements" for consideration at PLOS Computational Biology. As with all papers reviewed by the journal, your manuscript was reviewed by members of the editorial board and by several independent reviewers. The reviewers appreciated the attention to an important topic. Based on the reviews, I would like you to revise your manuscript and provide a clear indication in the main text/abstract about the precision and recall of your method.

Sincerely,

Anna R Panchenko

Associate Editor

PLOS Computational Biology

Florian Markowetz

Deputy Editor

PLOS Computational Biology

[LINK]

Reviewer's Responses to Questions

**Comments to the Authors:**

Reviewer #1: The authors have addressed my concerns

Reviewer #2: “Selecting the most important features at the individual level of cancer genome, for the first time we could build the model that outperforms all previous machine learning models for cancer breakpoint predictions”

###

Authors: “We included PR AUC values and updated the corresponding tables and figures: Figure S15 and Table S18 – combined models on full feature set for hotspots; …”

I appreciate the extra work the authors did to include precision and recall for their models and features. My main concern is now that these numbers are hidden away in supplemental tables, and nowhere in the manuscript is it mentioned that their models are only achieving 2-5% precision and recall <40%, resulting in an AUCPR <0.02 (e.g. Fig S15). Overall, this suggest that their models are good at predicting regions where there are NO breakpoints (negative class) but very poor at predicting actual breakpoint regions (the very rare positive class), resulting in high AUCs and very low AUCPRs. This is a classical machine learning problem and pitfall with class-imbalanced data. Based on the newly reported precision/recall data, I am concerned that this study will severely mislead the field with a claim that cancer breakpoints can be be robustly predicted from genomic/epigenomic features:

Abstract: "predictive models for cancer breakpoint formation achieved 70-90% ROC AUC for different types of cancers."

Introduction: “... for the first time we could build the model that outperforms all previous machine learning models for cancer breakpoint predictions”

I would recommend that the authors rework their manuscript and figures with a fair, unbiased and open discussion of the low precision and AUCPR.

**Have all data underlying the figures and results presented in the manuscript been provided?**

Reviewer #1: None

Reviewer #2: **No: **There is no code and data (eg. Python Notebook or Rmarkdown) that can reproduce the figures presented in the manuscript.

PLOS authors have the option to publish the peer review history of their article (what does this mean?). If published, this will include your full peer review and any attached files.

Reviewer #1: No

Reviewer #2: No
---

## [Editor Report · Decision Letter 3]

7 Jan 2021

Dear Dr. Poptsova,

Thank you very much for submitting your manuscript "Comprehensive analysis of cancer breakpoints reveals signatures of genetic and epigenetic contribution to cancer genome rearrangements" for consideration at PLOS Computational Biology. As with all papers reviewed by the journal, your manuscript was reviewed by members of the editorial board and by several independent reviewers.

I am ready to accept your paper but it needs to be proof-read to correct multiple English errors. Unfortunately our journal does not offer such services.

Sincerely,

Anna R Panchenko

Associate Editor

PLOS Computational Biology

Florian Markowetz

Deputy Editor

PLOS Computational Biology

[LINK]
---

## [Editor Report · Decision Letter 4]

28 Jan 2021

Dear Dr. Poptsova,

We are pleased to inform you that your manuscript 'Comprehensive analysis of cancer breakpoints reveals signatures of genetic and epigenetic contribution to cancer genome rearrangements' has been provisionally accepted for publication in PLOS Computational Biology.

I still see quite many grammatical errors in the manuscript, I suggest using Grammarly or other English editing software to fix the errors.

Best regards,

Anna R Panchenko

Associate Editor

PLOS Computational Biology

Florian Markowetz

Deputy Editor

PLOS Computational Biology

---

## [Editor Report · Acceptance letter]

18 Feb 2021

PCOMPBIOL-D-20-00990R4 

Comprehensive analysis of cancer breakpoints reveals signatures of genetic and epigenetic contribution to cancer genome rearrangements

Dear Dr Poptsova,

I am pleased to inform you that your manuscript has been formally accepted for publication in PLOS Computational Biology. Your manuscript is now with our production department and you will be notified of the publication date in due course.

With kind regards,

Andrea Szabo
